# *Bta06987*, Encoding a Peptide of the AKH/RPCH Family: A Role of Energy Mobilization in *Bemisia tabaci*

**DOI:** 10.3390/insects13090834

**Published:** 2022-09-13

**Authors:** Xiaofan Fan, Yong Liu, Zhuo Zhang, Zhanhong Zhang, Jing Peng, Yang Gao, Limin Zheng, Jianbin Chen, Jiao Du, Shuo Yan, Xuguo Zhou, Xiaobin Shi, Deyong Zhang

**Affiliations:** 1Longping Branch, College of Biology, Hunan University, Changsha 410125, China; 2Institute of Plant Protection, Hunan Academy of Agricultural Sciences, Changsha 410125, China; 3Institute of Vegetable Crops, Hunan Academy of Agricultural Sciences, Changsha 410125, China; 4Department of Entomology, University of Kentucky, Lexington, KY 40546, USA

**Keywords:** *Bta06987*, neuropeptide precursor, AKH/RPCH family, trehalose, RNAi, EPG, *Bemisia tabaci*

## Abstract

**Simple Summary:**

*Bta06987* gene encodes a peptide that belongs to the AKH/RPCH family. Members of this family exert essential effects on energy metabolism and other physiological and biological processes. *Bemisia tabaci* is a crop pest, causing crop losses mainly through feeding and transmission of plant viruses. There are only a few reports about AKH family in *B. tabaci*. In our study, we report on the characterization of the neuropeptide precursor encoded by *Bta06987* in *B. tabaci* at the molecular level and provide evidence for its essential function in energy homeostasis. Moreover, through an electrical penetration graph experiment, we found that manipulation of the relative expression of *Bta06987* effected the feeding behavior of *B. tabaci*. The findings in this paper can be used toward developing strategies aimed at decreasing the feeding time of *B. tabaci*, which might be useful for controlling pests and plant virus transmission.

**Abstract:**

A neuropeptide precursor encoded by *Bta06987* associates with AKH neuropeptide. In the AKH/RPCH family, these members have been demonstrated to participate in energy mobilization in many insects. In our research, the *Bta06987* gene from *Bemisia tabaci* was cloned, and the amino acid sequence analysis was performed. During the starvation of *B. tabaci*, the mRNA level of *Bta06987* showed a significant elevation. We investigated the functions of *Bta06987* in *B. tabaci* using RNA interference (RNAi), and the adult females of *B. tabaci* after being fed with *dsBta06987* showed a higher glycogen and triglyceride levels and lower trehalose content than the control. Furthermore, in the electrical penetration graph (EPG) experiment, *B. tabaci* showed changes in feeding behavior after feeding with *dsBta06987*, such as the reduction in parameters of E waveform percentage and total feeding time. Our findings might be helpful in developing strategies to control pest and plant virus transmission.

## 1. Introduction

Neuropeptides are known to regulate a diverse range of processes. These processes may include reproduction, development, metabolism, and behavior [1]. The adipokinetic hormone/red-pigment-concentrating hormone (AKH/RPCH) family is one of the essential groups of insect neuropeptides. Their members are produced by corpora cardiaca and exert effects on energy metabolism and other physiological and biological processes. The classical energy mobilization during energy-demanding activities is controlled by adipokinetic/hypertrehalosemic/hyperprolinemic hormones (AKHs) [2,3]. To date, more than 80 analogs of AKHs have been predicted or identified in different insects [4]. AKH peptides family is usually characterized by the following features: (I) position 4 and 8 have aromatic residues; (II) position 9 has a glycine residue; (III) an N-terminus modified by pyroglutamate and an amidated C-terminus; (IV) position 6, 7, and 10 always have varied amino acids; (V) length of 8–10 amino acids [5].

AKHs are multifunctional in various insects [6]. A primary role for AKHs is energy mobilization. It is thought that the regulation of energy balance is controlled by two systems. The one relies on glycogenolysis, where the AKHs bind to G protein-coupled receptors (GPCRs) and then activates protein kinase A (PKA). This process activates glycogen phosphorylase (GlyP), which is responsible for breaking down glycogen. The other system works via triacylglycerol lipase, which induces the mobilization of lipids in fat body [7,8].

AKHs generally exert their functions of energy metabolism on energy-demanding activities such as locomotion, development, and reproduction in insects [6,9]. In *Drosophila melanogaster*, ablating AKH-producing neuroendocrine cell decreased the elevation of trehalose in hemolymph and reduced locomotion significantly [10]. In addition, AKHs have been widely recognized as the triggering factor of catabolism in the fat body, which can fuel oogenesis in females insects [7]. In *Blattella germanica*, the suppression of the hypertrehalosemic hormone (HTH) gene expression by RNAi disturbed trehalose homeostasis and prolonged oviposition [11].

In order to maintain energy balance and homeostasis, the brain can regulate feeding behaviors according to alterations in internal metabolic states [12]. In *Acyrthosiphon pisum*, the neuropeptide F(NPF) is demonstrated to affect feeding behaviors and lead to a reduction in food intake [13]. In addition, NPF and other related peptides are shown to influence behavior such as food response and food-search behavior in adult files under the food deprivation condition [14,15]. Other neuropeptides, such as AKH, have been shown to affect feeding-related behaviors in *D. Melanogaster* and *A. pisum* [16,17].

*B. tabaci* is a kind of sap-feeding insect belonging to Hemiptera. It can suck nutrients from phloem sap in plant. The sap-feeding insect has critical implications in terms of the transmission of plant viruses [18,19]. For example, treating whiteflies with validamycin, which significantly influences the feeding behavior of *B. tabaci*, reduces the *Tomato Chlorosis Virus* (ToCV) transmission [20]. The feeding behaviors of sap-feeding insect are generally studied by the electrical penetration graph (EPG) technique. The EPG waveforms data reveal the details of insect probing activities for further study the interactions between sap-feeding insect and host plants [21,22].

Here, we found a *Bta06987* gene in whitefly database, which can encode a precursor of the AKH. Therefore, we hypothesize that the expression of *Bta06987* can influence energy metabolism and feeding behavior of *B. tabaci*. In order to confirm the role of *Bta06987*, we carried out following experiments. Firstly, we cloned the gene that encoded *Bta11975* and measured the relative gene expression of *Bta06987* after starvation for different times. Secondly, RNA interference (RNAi) was used to assess physiological function of *Bta06987* genes. Lastly, we identified the influence of *Bta06987* on the feeding behavior with the electrical penetration graph (EPG) technique. Our results might be helpful in controlling pest and plant virus transmission.

## 2. Materials and Methods

### 2.1. Plants and B. tabaci

The population of *B. tabaci* MED was obtained from Dr. Youjun Zhang’s research group at the Chinese Academy of Agricultural Sciences and reared on cotton plants in insect-proof cages in a greenhouse under the condition of 26  ±  2 °C, 75  ±  5% RH, and long-day photoperiods (14 h light: 10 h dark). In order to confirm the purity of the *B. tabaci* colony, 50 whiteflies were randomly collected every 60 days for identification of the *B. tabaci* biotype [23]. In the experiment, insects were newly emerged females and all whiteflies in our research were virus-free.

Seeds of tomato (*Solanum lycopersicum* Mill. Cv. Zuanhongmeina) were sown in nutritional soil on a plastic plate in the greenhouse under the condition of 26  ±  2 °C, 75 ±  5% RH, and long-day photoperiods (14 h light: 10 h dark). The seedlings were transplanted into plastic pots in insect-proof cages until they had grown 2–3 true leaves. After 15 days, the plant under the same growth form could be used for EPG.

### 2.2. Gene Cloning

Total RNA extracted from whiteflies was used as starting material and the reverse transcription was performed to obtain cDNA. Basing on the DNA sequence of *Bta11975* gene in whitefly database, the primers were synthesized using Primer 5 [24] (Table 1). The cDNA of whiteflies was regarded as a template to amplify the target gene. PCR was the performed using the Phanta^®^ Max Super-Fidelity DNA Polymerase Kit (Vazyme Biology Co., Ltd., Nanjing, China). The PCR-amplified products were assessed via electrophoresis on 2% agarose gels. After sequencing and purification, the target DNA fragments were ligated into the pEASY-Blunt cloning vector and the recombinations were transferred into Escherichia coli Trans1-T1 competent cells to acquire the transformants, which were uniformly inoculated onto ampicillin-containing LB medium plates for culture at 37 °C overnight. Then, single colonies were picked from the LB plate, and the positive recombinants were screened by PCR. Then, the positive recombinants were inoculated into ampicillin-containing liquid LB medium for amplification, and we could get the clone plasmids. The cloned plasmids were sent to company for sequencing analysis and the fragment of *Bta06987* was obtained.

### 2.3. Characterization and Phylogenetic Analysis

The amino acid sequences of AKHs were acquired from Whitefly Genome Database (http://www.whiteflygenomics.org/cgi-bin/bta/index.cgi, access on 8 October 2021) and GenBank. The precursor encoded by *Bta06987* included the predicted signal peptide (http://www.cbs.dtu.dk/services/SignalP/, access on 8 October 2021) and the prediction of mature peptide. The phylogenetic tree was constructed using neighbor-joining method (bootstrapping with 1000 replicates) under Clustal X2 [25] and MEGA 7.0 [26]. The evolutionary position of *Bta06987* was ascertained by comparison of the sequences with other insect prohormones of AKHs.

### 2.4. The relative Expression of Bta06987 under Starvation

In this study, quantitative real-time PCR (qPCR) was used to quantify relative gene expression. The total RNA of whiteflies was extracted by TRI reagent and the reverse transcription was performed using Hiscript^®^ II Q RT SuperMix for qPCR Kit (+g DNA wiper); Fluorescence quantitative PCR was conducted with Cham Q Universal SYBR qPCR Master Mix Kit (TransGen Biotech. Co., Ltd., Beijing, China). Specific primers used in this assay were listed in Table 1. The actin genes (LOC109031276) served as an endogenous reference gene and the relative expression of genes was analyzed by 2^−ΔΔCt^ method. All of the qPCR experiments were performed in three biological replicates. For the starvation research, newly emerged female whiteflies were placed in glass tubes (Appendix A) (50 whiteflies/tube). The females and males are distinguished by a microscope. After 2, 4, 6, and 8 h, those centrifugal tubes were placed into liquid nitrogen for quick-freezing. In the refed study, whiteflies that had been starved for 6 h were placed on cotton plants. After another 3 h, whiteflies were placed in 1.5 mL centrifugal tubes and then put into liquid nitrogen for quick-freezing. Genes expression was quantified by qPCR.

### 2.5. Double-Stranded RNA Synthesis

Following the manufacturer’s protocol, the corresponding dsRNA was synthesized using the T7 RiboMAX Express RNAi system of the dsRNA synthesis kit (Promega, Madison, WI, USA). *Bta06987*fragments for dsRNA synthesis were amplified by specific primers containing the T7 promoter sequence (Table 1). The primers are designed by Primer 5 basing on the sequence of *Bta06987* (Appendix A). After purification using the cycle pure kit (Omega Biotek Inc., GA, USA), the PCR product was mixed with T7 Express Enzyme Mix and RiboMAX™ Express T7 2 × Buffer at a specific proportion. The mixture was incubated at 37 °C for 2–6 h and at 70 °C for 10 min, and the mixture was then transferred for incubation at room temperature for 20 min to promote dsRNA formation. To finish, we added DNase and RNase A into dsRNA to remove DNA template and single-stranded RNA, respectively. The quality of dsRNA was determined on a 2% agarose gel and the quantity of dsRNA was measured by Nanodrop 2000.

### 2.6. Efficacy of dsRNA

Next, the dsRNA was mixed with 15% (*w*/*v*) sucrose solution in the proportions required to obtain desired concentrations. Parafilm was extended as thin as possible to seal one end of the transparent glass tube. The corresponding concentrations of dsRNA were added on the Parafilm and another Parafilm was used to cover it. Next, 100 whiteflies were placed in the glass tube and then a black sponge was used to clog the other end of glass tube. Lastly, silver paper was used to cover glass tube (Appendix A). Those glass tubes were placed in a greenhouse (26 ± 2 °C, 75 ± 5% RH, and long-day photoperiods (16 h light: 8 h dark)). According to the previous RNAi feeding experiments in *B. tabaci* [27,28], 400 ng μL^−1^
*Bta06987* dsRNA was chosen for feeding the whiteflies to analyze the efficiency of RNA interference. Then, we compared the effects of RNAi with different concentrations (400 and 500ng μL^−1^) for three different times (48 h, 72 h, 96 h) to select the optimal efficiency.

### 2.7. Carbohydrate Content and Triacylglycerol (TAG) Content Determination after RNAi

A total of 70 whiteflies were collected from transparent glass tubes into a 1.5 mL centrifuge tube to be frozen by liquid nitrogen and then stored at −80 °C. The biomass was calculated in each case based on three biological replicates. The whiteflies placed in centrifuge tube were ground using a pestle, and the corresponding extraction liquids were then mixed with whiteflies according to the manufacturer’s instruction. The contents of glycogen, trehalose and glucose were determined using the corresponding assay kit. The trehalase activity was measured based on 3,5-dinitrosalicylic acid method using the Trehalase Kit. The content of triglyceride was measured using the Triglyceride Kit. All kits were purchased from Solebo Technology Co., Ltd. (Beijing, China).

### 2.8. Feeding Behavior after RNAi

Whiteflies were placed in Duchenne tubular. There was only one whitefly in one Duchenne tubular. After starving for 2 h, the whiteflies that had been fed with dsRNA were attached to a gold wire electrode (2 cm × 12 μm) using electrically conductive silver glue. The electrodes were connected to an eight-channel direct-current EPG system. The EPG data was recorded with Stylet + for Windows software (EPG stylet + d) [29]. The feeding behavior was recorded continuously for 8 h. All recording experiments were carried out inside a grounded Faraday cage in a quiet room (26 ± 2 °C, 75% ± 5% RH). Based on the previous description, EPG waveforms were categorized into four groups: np (non-probing behavior), C (showing insect intercellular stylet pathway), E1 (phloem salivation), and E2 (phloem ingestion) [30].

### 2.9. Statistical Analysis

The data in the figures are presented as the mean ± SEM using SPSS 25.0 (SPSS Inc., Chicago, IL, USA). Statistical differences in the study of the relative expression of *Bta06987* under starvation were analyzed by Student’s *t*-tests (Figure 2). Statistical differences in the relative expression of *Bta06987* after RNAi and *B. tabaci* mortality were analyzed by one-way analysis of variance (ANOVA) followed by Duncan’s multiple range test (Duncan’s MRT) (Figure 3); Statistical differences in the carbohydrate content and triglyceride content were analyzed by Student’s *t*-tests (Figures 4 and 5); Statistical differences in the stylet activities were nalyzed by one-way ANOVA and Duncan’s MRT (Figure 6). *p* < 0.05 was regarded as indicating statistically significant differences.

## 3. Results

### 3.1. Cloning and Phylogenetic Analysis of Bta06987

Cloning the gene of *Bta06987* was completed by the RT-PCR technique. The length of *Bta06987* we obtained was 209 bp (Appendix A). *Bta06987* encoded a predicted 71 amino acid AKH neuropeptide precursors (Appendix A). In the phylogenetic tree, we compared the amino acid sequences of AKHs in different insects. Our result showed that *Bta06987* appeared in the same clade with other known AKHs. In addition, it was strongly associated with *L. striatella* and *N. lugens* AKH. These proteins closely clustered, which suggested that the peptide encode by *Bta06987* may function similarly to AKH in other insects (Figure 1).

### 3.2. The Expression of Bta06987 under Starvation

The relative expression level of the *Bta06987* under starvation was analyzed using the qPCR techniques. Starvation led to a marked increase in *Bta06987* expression. In contrast to fed females, the mRNA expression levels of *Bta06987* were elevated 1.45-, 2.01-, 2.71-, and 2.34- fold after 2, 4, 6, and 8 h of starvation, respectively (Figure 2A). Furthermore, a food rescue study was performed on the starved female adults (starvation for 6 h followed by refeeding for 3 h). The supplement of food after famine led to an obvious reduction in *Bta06987* mRNA transcript levels (Figure 2B).

### 3.3. Efficacy of dsRNA

In order to estimate the effects of RNAi of the *Bta06987* gene, feeding assays with different concentrations of *dsBta06987* (400 ng μL^−1^, 500 ng μL^−1^) were conducted at different times in whiteflies. In addition, the mortality rates were evaluated at 0, 24, 48, 72, 96, and 120 h. The qPCR results showed that the *Bta06987* mRNA expression level of *dsBta06987* group significantly reduced from 48 to 96 h compared to double-stranded green fluorescent protein group (*dsGFP*). Feeding with 500 ng μL^−1^
*dsBta06987* dramatically reduced *Bta06987* expression in 72 h compared with feeding of 400 ng μL^−1^
*dsBta06987* (Figure 3A). The whitefly mortality, after feeding with *dsBta06987*, gradually increased with increasing time. However, there was no significant difference in feeding with *dsGFP* and *dsBta06987* for 400 ng μL^−1^ or 500 ng μL^−1^ (Figure 3B). Therefore, 500ng μL^−1^ dsRNA and 72 h RNAi were selected for the subsequent test.

### 3.4. Carbohydrate Content

Compared with the control, female adults fed with *dsBta06987* demonstrated a decrease in the trehalose content by 29% (Figure 4A). The mRNA level of the trehalase 1 (*Tre-1*) gene and trehalase 2 (*Tre-2*) in females increased significantly after *dsBta06987* feeding (Figure 4B,C). As a result, the activity of trehalase increased dramatically (Figure 4D). In addition, feeding with synthetic *dsBta06987*significantly elevated glycogen and glucose content by 1.6- and 1.3-fold, respectively, compared to the levels in females fed with *dsGFP* (Figure 4E,F). However, there is no significant difference in the expression of trehalose-6-phosphate synthase 1 (*TPS1*) and trehalose-6-phosphate synthase 2 (*TPS2*) between dsGFP and *dsBta06987* (Figure 4G,H).

### 3.5. Triglyceride Content

We then identified whether feeding with *dsBta06987* affected the lipid mobilization in *B. tabaci*. As shown in Figure 5A, feeding with synthetic *dsBta06987* significantly increased the content of TAG by 1.7-fold when compared to feeding with *dsGFP.* Moreover, the relative expression of hormone-sensitive triglyceride lipase (*HSL*) decreased dramatically in *dsBta06987*-fed group.

### 3.6. Stylet Activities

Compared to the group fed with *dsGFP*, the total duration of probes (F_1, 15_  =  6.433, *p*  <  0.05) and total time of waveforms E1 and E2 (E1: F_1, 15_  =  7.718, *p*  <  0.01; E2: F_1, 15_ = 10.433, *p*  <  0.01) of whiteflies fed with *dsBta06987* dramatically decrease (Figure 6A,F,H); The number of probes (F_1, 15_  =  17.615, *p*  <  0.01) markedly increased, and the time from the first probe to the first E (pd) (F_1, 15_  =  13.096, *p*  <  0.01) was significantly longer in the *dsBta06987* group (Figure 6B,D). However, no significant differences were found between *dsGFP* and *dsBta06987* in the total duration of waveform C (F_1, 15_  =  0.219, *p* >  0.05, Figure 6C) and the number of E1 and E2 waveforms. (E1: F_1, 15_  =  2.922, *p* >  0.05; E2: F_1, 15_  =  1.715, *p* >  0.05, Figure 6E,G). Our study indicated that *Bta06987* interrupted the feeding behavior of whiteflies.

## 4. Discussion

In the AKH/RPCH family, the report by Steele is the first on a hyperglycemic factor, which was from the American cockroach (*Periplaneta americana*) [31], and since then, about 80 AKHs have been discovered in insects. These neuropeptides have been thoroughly reviewed regarding their series of important physiological functions such as energy homeostasis, reproduction, and immunity [32]. However, there are only a few reports about AKH family in *B. tabaci* [4,33]. In the present study, we report the characterization of precursor of the AKH encoding by *Bta06987* at the molecular level (Appendix A) and construct a phylogenetic tree (Figure 1). The analysis of the predicted amino acid sequences indicates that *Bta06987* harbor a conserved domain, which belongs to the adipokinetic hormone super family. Those result may suggest that *Bta06987* has an important function in energy balance.

Energy homeostasis is necessary for organisms to ensure proper growth and adaption to the environment. Insulin and glucagon signaling are the crucial mechanisms for maintaining energy balance in mammals. Under the condition of rich nutrition, the insulin that secreted by pancreatic cells binds to the insulin receptor to activate the Phosphatidylinositol-3-Kinase (PI3K) and the mitogen-activated Protein Kinase (MAP-Kinase). Activation of the PI3K pathway can stimulate glucose uptake and promote energy storage. Under increasing energy demand and limited nutrition conditions, glucagon signals promote lipids and glycogen mobilization via G protein coupled receptor (GPCR) to meet ongoing energy needs. Similarly to mammals, insect energy homeostasis is reliant on the AKHs (a functional glucagon homolog) under starvation conditions [34]. In our study, we found that the relative expression of *Bta06987* increased significantly during starvation (Figure 2). A previous study on cockroach *Blaberus discoidalis* suggested that food deprivation stimulated the secretion of hypertrehalosemic hormones, which was regarded as an adaptation to starvation [35]. The hungry flies showed hyperactive foraging activity, whereas the neurons with AKH loss or AKHR inactivation suppressed this behavior [36]. These results suggested that AKHs may promote energy mobilization and maintain high activity to enable insects to find food under starvation conditions. In the refed investigation, we found that the relative expression of *Bta06987* in refed group decreased significantly compared to the starvation treatment (Figure 2B). This result indicated the mRNA level of *Bta06987* is closely linked with insects’ own energy balance. In addition, the result of mortality of whiteflies in RNAi showed no significant difference in feeding with *dsGFP* and *dsBta06987* (Figure 2B). The half-life of the *B.tabaci* is about 26 days in April [37], which is much longer than 120 h. This might suggest that RNAi did not cause mortality by decreasing the relative expression level of *Bta06987* or *Bta06987*, which is not indispensable for survival. In insects, the fat body is functional equivalent to vertebrate liver and adipose tissues, and it is the crucial organ for nutrient storage organ in insects. Stored energy is mainly in the form of glycogen and triglycerides [38]. In the *D. melanogaster* and *Periplaneta Americana*, AKHs can derive glucose into trehalose synthesis in the fat body [39,40]. Trehalose, a non-reducing disaccharide with two glucose molecules, is prevalent and ubiquitous in the biological world, but it has not been found in mammals. As the instant source of energy, trehalose, also known as “blood sugar”, is involved in insect development and adaptation to abiotic stresses [41]. Our results demonstrated that *Bta06987* signaling pathway is important for trehalose homeostasis. In our study, the reduction of *Bta06987* expression was found to decrease the content of glycogen (Figure 4E), which showed that *Bta06987* might regulate trehalose content by influencing the glycogenolysis. More importantly, the trehalose content also seems to be controlled by the activity of two enzymes. One is trehalose-6-phosphate (*TPS*), which can catalyze trehalose synthesis, and the other is called trehalase, which can convert the trehalose into glucose [42,43,44]. Here, our results demonstrate that RNAi of *Bta06987* significantly increased the expression of two *TPS* genes in *B. tabaci*, however, there were no significant differences in *TPS1* and *TPS2* between the treatments (Figure 4). These results suggested that the regulation of trehalose abundance may be controlled by *TRE*, as in *N. lugens* [45]. In addition, the increase in glucose content provides further support for this idea (Figure 4). Notably, triacylglycerols (TAGs) also are the primary energy stored in the body [46]. Depending on the function of AKHs signaling system to regulate lipid in other insects [47,48], we assume that *Bta06987* might play a role in modulating lipid catabolism. In insects, lipids are stored primarily in the form of triglycerides [49]. As presented in the result, the TAG content increased significantly (Figure 5A), and the relative expression of hormone-sensitive triglyceride lipase (*HSL*), which is one of the major enzymes contributing to TAG breakdown, decreased dramatically. The founding indicated the role of *Bta06987* in regulating lipids.

Another important instance of AKHs functions in evolutionary divergence concerns feeding behaviors [50]. Compared to energy mobilization, the role of feeding behavior in AKHs is relatively underexplored. In previous research, AKH signaling reduced feeding in cricket *Gryllus bimaculatus* and it was also an orexigenic peptide in *D. melanogaster* [50,51]. In our study, considering the distinct down-regulation of *Bta06987* after RNAi experiment(Figure 2A), we investigated the possible function of *Bta06987* in feeding using RNAi technique. We analyzed the change in feeding behavior in *B. tabaci* after feeding with dsRNA using electrical penetration graph (EPG) technique. We found the RNAi of *Bta06987* markedly decreased insect food intake. The total duration of C and E waveform decreased significantly (Figure 6), and the proportion of C and E waveform dramatically reduce in eight hours (Figure 7). Moreover, the data of E waveform indicated the priority of insects for phloem sap [52], suggesting the delay of phloem feeding by *B. tabaci* is due to their reduced appetite. The reduction of food intake in turn reduced the direct damage to plants caused by stylet probing. Besides, the duration time of E waveform is closely linked to virus transmission [53,54], which indicated that silencing of *Bta06987* may suggest a new way to reduce the transmission of plant virus. Therefore, the insect neuropeptides of AKH family may emerge as essential targets in pest management. However, it should be noted that the physiological basis of appetite loss caused by the absence of AKHs is unclear. In *D. melanogaster*, AKHs influenced the expression of neuropeptide F, which regulated feeding [55]. Additionally, recent studies also showed that feeding behavior is regulated by trehalose and glucose levels [27,56]. Thus, the true mechanism of action of AKH family to regulate feeding behavior needs to be studied further.

## 5. Conclusions

In this work, we cloned and performed characterization and phylogenetic analysis of *Bta06987* in *B. tabaci*. We found the relative expression of *Bta06987* increased significantly under starvation, which indicated that *Bta06987* may affect energy mobilization for insect to seek food. As a result, by RNAi, we researched that *Bta06987* played an essential role in energy homeostasis, namely through regulating energy mobilization via lipolysis and glycogenolysis and, more importantly, influencing the activity of trehalase, which catabolizes trehalose. Moreover, in the electrical penetration graph experiment, we found that *Bta06987* gene deletion effected the feeding behavior of *B. tabaci*. Thus, a strategy based on appetite loss and decreasing feeding time in phloem might be helpful in controlling pest and plant virus transmission.

## Figures and Tables

**Figure 1 insects-13-00834-f001:**
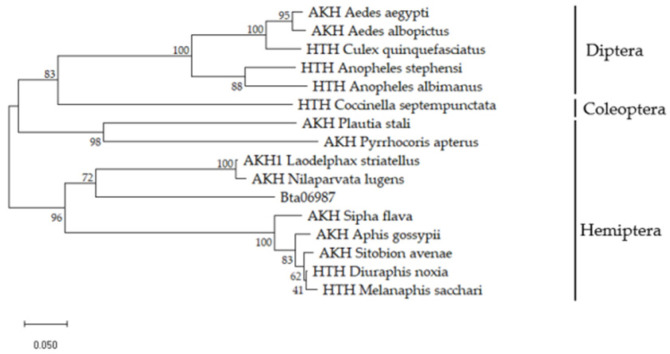
A phylogenetic tree of AKHs sequences. The tree was constructed by Clustal X2 and MEGA 7 with 1000 bootstrap replicates. The bar size indicates the number of amino acid substitutions per site. The GenBank accession numbers of insects AKHs are as follows: *Ae. aegypti* (CAY77165.1); *Ae. Albopictus* (JAV47133.1); *Cx. quinquefasciatus* (XP_001842492.1); *An. Stephensi* (XP_035915439.1); *An. albimanus* (XP_035775852.1); *C. septempunctata* (XP_044765111.1); *P. stali* (BAV78787.1); *P. apterus* (AGZ62588.1); *L. striatella* (AXF48182.1); *N.lugens* (AFN26934.1); *S. flava* (XP_025422856.1); *A.gossypii* (XP_027850481.1); *S. avenae* (ALH44120.1); *D. noxia* (XP_015374547.1); *M. sacchari* (XP_025194429.1).

**Figure 2 insects-13-00834-f002:**
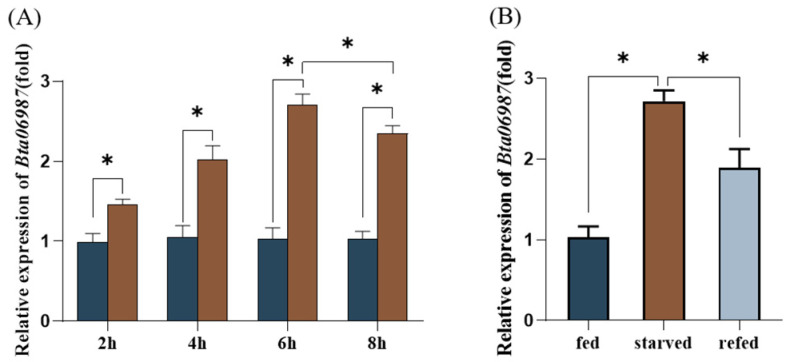
*Bta06987* relative expression after starvation. (**A**) Whiteflies were placed in 1.5 mL centrifugal tubes for starvation with 2, 4, 6, and 8 h. (**B**) Food rescue experiment. Data were shown as the mean ± SEM and evaluated by *t*-test. Bars labeled with the ‘*’ denote significant differences at *p* < 0.05.

**Figure 3 insects-13-00834-f003:**
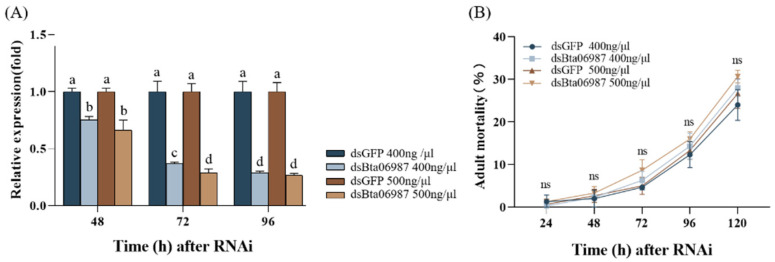
Effect of different doses of dsRNA in (**A**) the relative gene expression of *Bta06987* with different time in whiteflies and on (**B**) the mortality of whiteflies with different time. RNAi was performed in 48 h, 72 h, and 96 h with 400 ng μL^−1^ and 500 ng μL^−1^.The mortality was estimated every 24 h from 24 to 120 h. All data are presented as the mean ± SEM and evaluated by one-way ANOVA analysis and Duncan’s multiple-range test. Different letters on the histogram indicated significant differences at *p* < 0.05. The ‘ns’ on the line indicated on significant difference at *p* > 0.05.

**Figure 4 insects-13-00834-f004:**
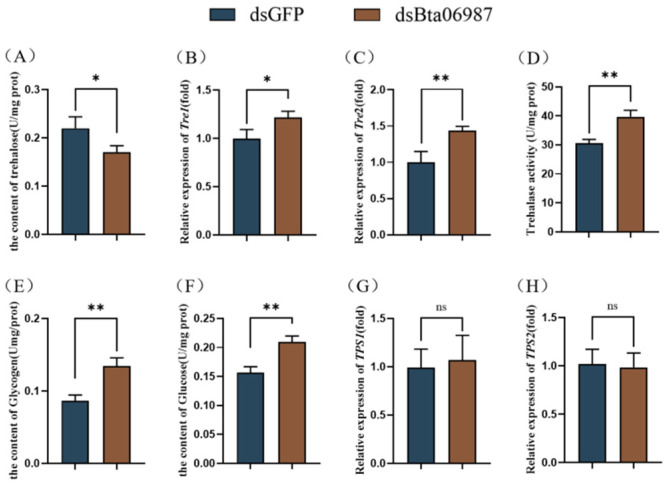
Carbohydrate content and the relative expression of related genes. (**A**) The content of trehalose and (**B**) the relative expression of *Tre1*. (**C**) *Tre2*, (**D**) the activity of trehalase, (**E**) the content of glycogen, (**F**) the content of glucose. (**G**) The relative expression of *TPS1* and (**H**) *TPS2*. The data are shown as means ± SE of three independent experiments. Significant differences based on Student’s *t*-test are denoted as * *p*  <  0.05 and ** *p*  <  0.01; no significant differences based on Student’s *t*-test are denoted as ns.

**Figure 5 insects-13-00834-f005:**
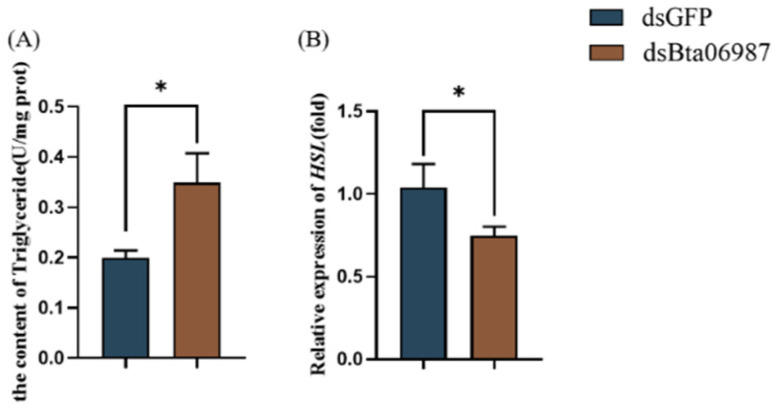
Triglyceride content and the relative expression of related gene. (**A**) The content of TAG. (**B**) The relative expression of *HSL*. The data are shown as means ± SE of three independent experiments. Significant differences based on Student’s *t*-test are denoted as * *p*  <  0.05.

**Figure 6 insects-13-00834-f006:**
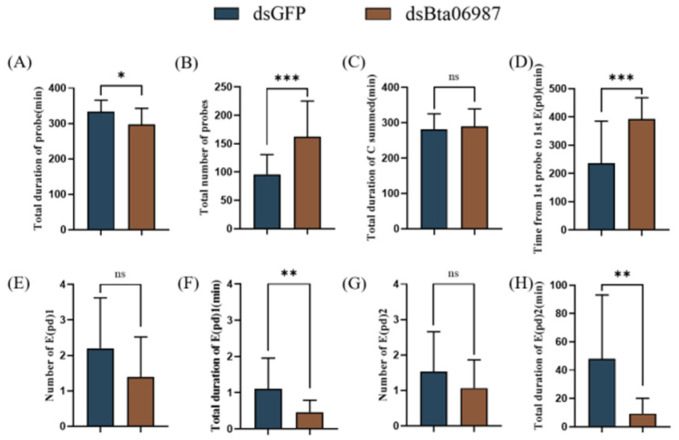
Feeding behavior of whiteflies. (**A**): Total duration of probes; (**B**): total number of probes; (**C**): total duration of C; (**D**): time from the first probe to the first E(pd); (**E)**: number of E(pd)1; (**F**): total duration of E(pd)1; (**G**): number of E(pd)2; (**H**): total duration of E(pd)2. The data are shown as means ± SE with 15 independent experiments. Significant differences based on repeated measures ANOVA are denoted as * *p* < 0.05, ** *p* < 0.01, and *** *p* < 0.001. No significant differences based on Student’s *t*-test are denoted as ns.

**Figure 7 insects-13-00834-f007:**
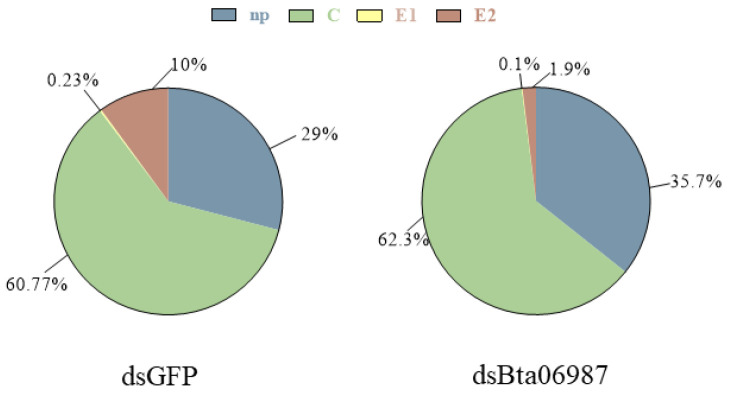
Mean percentage of the EPG waveforms in 8 h feeding. dsGFP: *B. tabaci* fed with dsRNA of *GFP*, dsBta06987: *B. tabaci* fed with dsRNA of *Bta06987.* The waveform for intercellular stylet pathway (C) and non-probing (np), waveform for salivation into phloem sieve elements (E1), and phloem ingestion (E2).

**Table 1 insects-13-00834-t001:** Primers used in the study.

Primers	Primer Sequence (5′-3′)
**For RT-PCR**	
*Bta06987*-F	CTTGTCGCACAATTCTGGTGT
*Bta06987*-R	ACTTCTGAACTTCTCACAATC
**For qPCR**	
*Bta06987*-F	AGACAATCCTCCTGTCCGCTCTG
*Bta06987*-R	ACTTCTGAACTTCTCACAATC
THL1-F	TTGGAGGCGGAGGTGAG
THL1-R	GCAGTTAGTTTGGGAGGGG
THL2-F	ACGGATGACGGCAGAAGA
THL2-R	ATGGCAAAAGGAGGAAAGG
TPS1-F	GCCCCTTTATTTTCCACCC
TPS1-R	AGATTCGTACATCCTATGCCTGTT
TPS2-F	AATGCGGGGTTCTCGTTTG
TPS2-R	GGTATCGCCATCACCTTCC
HSL-F	ACCTCAGAGCTGTTTAGTGCC
HSL-R	GAGGTGCAACTTTTCCTGGC
Actin-R	TCTTCCAGCCATCCTTCTTG
Actin-F	CGGTGATTTCCTTCTGCATT
**For dsRNA synthesis**	
*dsBta06987*-F	TAATACGACTCACTATAGGGCTTGTCGCACAATTCTGGTGT
*dsBta06987*-R	TAATACGACTCACTATAGGGAGATTGCGCCTCATTCTCGATCA
*dsGFP*-F	TAATACGACTCACTATAGGGTTCAGTGGAGAGGGTGAAGGT
*dsGFP*-R	TAATACGACTCACTATAGGGTGTGTGGACAGGTAATGGTTG

## Data Availability

Data is contained within the article.

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
