# Peer review of "Bta06987, Encoding a Peptide of the AKH/RPCH Family: A Role of Energy Mobilization in Bemisia tabaci"

_insects, 2022, doi:10.3390/insects13090834_

Round 1
Reviewer 1 Report (Previous Reviewer 1)
The undelined stntences are too many, so I could not read this MS easily. This MS looks a new MS, not revised version.
However, as a new MS, this MS can be accepted for publication after s small correction.
1. L73, "studied"?
2. L278 and 279, What is the diffrence between "feeding on " and "feeding with"?
3. L319, Only the sequence homology is not enough for indicating the function. You should correct to "may suggest".
Author Response
Response to Reviewer 1 Comments
Dear reviewer:
We would like to thank you for your careful reading, helpful comments, and constructive suggestions, which has significantly improved the presentation of our manuscript (ID: insects-1856930).We have carefully considered all comments from the reviewers and revised our manuscript accordingly. The manuscript has also been double-checked, and the typos and grammar errors we found have been corrected. In the following section, we summarize our responses to each comment.
Point 1: L73, "studied"?
Response 1: Done.(Line 74)
Point 2: L278 and 279, What is the diffrence between "feeding on " and "feeding with"?
Response 2: There is no different between "feeding on " and "feeding with". It is correctted in thw paper.(Line 288)
Point 3: L319, Only the sequence homology is not enough for indicating the function. You should correct to "may suggest".
Response 3: Done. (Line 324)
Reviewer 2 Report (Previous Reviewer 2)
Minor revision as follows-
1. What is Bta11975?
2. Primer 3.0 or Primer 5.0 (please provide reference or web address)
3. How sequence confirmation was performed? (Line no. 111)
4. Clustal X2 (Reference required)
5. How did you manually edit using the GeneDoc program? It is only a sequence visualization program. (Line no. 120)
6. Whitefly Genome Database (Reference or web address required)
7. Primer 3.0 or Primer 5.0 (Line no. 149)
8. Did you measure quantity of dsRNA? (Line no. 155)

Author Response
Response to Reviewer 2 Comments
Dear reviewer:
We would like to thank you for your careful reading, helpful comments, and constructive suggestions, which has significantly improved the presentation of our manuscript(ID: insects-1856930). We have carefully considered all comments from the reviewers and revised our manuscript accordingly. The manuscript has also been double-checked, and the typos and grammar errors we found have been corrected. In the following section, we summarize our responses to each comment.
Point 1: 1. What is Bta11975?
Response 1: It is a gene number we found in the Whitefly Genome Database. It has a conserved domain which belongs to the adipokinetic hormone super family.
Point 2 : Primer 3.0 or Primer 5.0 (please provide reference or web address)
Response 2: Done.(Line 103)
Point 3: How sequence confirmation was performed? (Line no. 111)
Response 3: It is add in the Line 111.
Point 4. Clustal X2 (Reference required)
Response 4: Done. (Line 120)
Point 5: How did you manually edit using the GeneDoc program? It is only a sequence visualization program. (Line no. 120)
Response 5: GeneDoc is an auxiliary software for protein and DNA sequence comparison. It can beautify and edit the results of other software.
Point 6: Whitefly Genome Database (Reference or web address required)
Response 6:Done. (Line 118)
Ponit 7: Primer 3.0 or Primer 5.0 (Line no. 149)
Response 7: Done. (Line 150)
Ponit 8: Did you measure quantity of dsRNA? (Line no. 155)
Response 8: It is add in the Line 157.
Reviewer 3 Report (Previous Reviewer 3)
Authors have tried to improve manuscript and have achieved that at certain places BUT in general there are still open questions and the English is still unacceptable. This referee has only checked now carefully the Title, simpl. e summary, abstract and introduction and has the following to criticize:
Title: should be "...of the AKH/RPCH ... Thus, incorrect English. Furthermore, authors should be a bit more cautious and rather say " A possible or a putative role ..."
Simple Summary: line 14: reports of the AKH family ...This statement is NOT true. In my last report I told authors already that there is a seminal work on Hemiptera AKHs published in Frontiers in Insect Sciences by Gaede and Marco. Of course, AKH of Bemisia is mentioned there.
line 14/15: In this study, we reported .. English grammar. Should be present tense
line 20: what does this mean [AH1]?
Abstract: line 23: English again: not sequences but singular
line 37: mistake: cardiaca
line 39: English grammar; not is control but is controlled
line 40: ...more than 60 ... Rather change to more than 80; this would be clear when authors had read the Hemiptera AKH publication in Frontiers in Insect Sciences
line 48: English grammar; not AKHs binds to .... but AKHs bind to
line 53: English. not ..energy-demand... but energy-demanding
line 58/59: English. not ...hypertrehalose hormone.... but hypertrehalosemic hormone
at many places: why is Drosophila given only a genus but all other insects are given genus plus species?
Fig. 1: legend does not state which insects belong to the order Hemiptera. Mind that there are MANY more sequences as shown in the Hemiptera paper in Frontiers in Insect Sciences which could be used. Why Coccinella was used as outgroup is still enigmatic (and in my view WRONG) especially when the sequence is so similar.
Also be aware of the current nomenclature of AKHs: 5 letter code
I leave it as this. Authors will see that the manuscript is FAR from ready.
Round 2
Reviewer 3 Report (Previous Reviewer 3)
There will be no comments to authors. I refuse to review a manuscript of a third time when I see that TEN authors have just copied my suggestions from 2nd round which dealt only with abstract and Introduction but have not done much work on the rest of the manuscript to improve English languag and understanding
Author Response
Thanks for you advises.
This manuscript is a resubmission of an earlier submission. The following is a list of the peer review reports and author responses from that submission.
Round 1
Reviewer 1 Report
First you should read instructions for authors of this journal very carefully. In addition, you should refer many insect neuropeptide papers as basical terms are inappropriate.
1. Species name should be in iitalic. Some are not in italic.
2. In introduction, Some are in past tense, but others are present.
3. in 2.4, which sequences were used for dsRNA are not shown. These data should be shown in supplemental data.
4. in Fig.1, signal peptide and clevage site are not shown.
5. in 3.3, you did not indicate which figure or Table show these data.
6. in Fig.3, you injected dsAKH, but you measured HTH expression. What does it mean?
7. in L216, "in addition" should be deleted. There is no meaning.
8. in L287-, in case of neuropeptide, we usually check if the sequence include "signal peptide" as the secretory peptide has such motief. However, you use "TM", but not "signal peptide" What does it mean?
9. in L288, I cannot agree to this hypothesis. Many secretory peptides/proteins have "signal peptide" motief. However, signal peptide does not cause the crusical function.
10. In your result, there was no significant difference in the mortality between dsGFP-injected insects and dsHTH injected insects (Fig.3). So it is unclear whether HTH has crusical function on survival even if HTH change the content of lipid and feeding behavior. HTH may have some functions but not crusical function.
11. However, the result of Fig.6 and 7 are interesting and you reconstruct the story to emphasis these results.
Reviewer 2 Report
Fan et al. have addressed the requirement of Hypertrehalosomic hormone (HTH) during starvation in Bemisia tabaci using RT-qPCR and RNAi approaches. The BmHTH silenced individuals showed changes in feeding behavior such as reduction of E waveform percentage and total feeding time. The study is important in the context of pest control and plant-virus transmission but major revisions need to be made before it can be published. The scientific language and English structure is poor at many places and need interventions from English language readers. Please complete the revisions to my comments as follows:
1. Cloning of HCH---> Present in detail. This section should come before phylogenetic analysis section.
2. I recommend Clustal X2 for phylogenetic analysis.
3. Sentence making and word usage needs to be corrected at many places.
4. Please re-write:
The phylogenetic tree was constructed using neighbor-joining method (bootstrapping with 1000 replicates) under MEGA 7.0 (pl. include citation??). The evolutionary position of BtHTH was ascertained by comparison of the sequence with other insect AKHs, keeping Coccinella septempunctata AKH as an outgroup.
5. Please re-write
The relative gene expression level of BtHCH was analyzed using qRT-PCR.
6. counted ---> analyzed (line no. 109)
7. performed in three biological replicates (line no. 110)
8. What is the endogenous control in qRT-PCR experiments?
9. Please mention regarding primer designing for dsRNA synthesis.
10. Elaborate on dsRNA synthesis and RNAi in the M&M section.
11. Line 141-144- According to previous study---- (Reference??) optimal efficiency of HTH dsRNA for gene silencing was studied previously.
12. Line 149-156- Please reduce redundancy in sentences.
13. Please mention BtHTH throughout the manuscript?
14. Scientific name in italics throughout the manuscript.
15. Please show the figure depicting BtHTH cDNA (216 bp)
16. Figure 4- Figure legends do not match the labels in figures. For eg. A. content of trehalose and not content of glycogen
Reviewer 3 Report
The Introduction is quite weak: authors start to call this peptide a hypertrehalosemic hormone but without any logic. Why is it not an adipokinetic hormone?
In their result section they even see that it
> has an influence on the fat metabolism. Thus calling this peptide
> hypertrehalosemic is not correct. The literatur used in the
> Introduction is partially very subjective. And in general one asks
> why the authors cite such a lot of papers on cockroaches and on
> Drosophila, although there are about 20 to 30 published work on
> actions of an AKH in various Hemiptera - and that is the group that
> Bemisia belongs to. Lines 66 to 68 in the Intro are also quite
> revealing: why stating again the general stuff of AKH family? Done
> before!
At places the English is not clear, this is especially clear in the
> M/M section. Some examples: the whole paragraph 2.5 is difficult to
> comprehend let alone to repeat. What does it mean "douse with
> sucrose solution" in line 135? In line 156 authors talk about a
> lipase kit with which they measured lipase activity. There are a
> number of totally different lipases. Authors must be much more
> specific what exactly they have measured here and how. Line 158: no
> one will understand this sentence: "Whiteflies were placed in Duchenne
> tubular one by one".
The phylogenetic analysis of Bt HTH is a joke:
> why where the AKHs from Diptera, Blattodea etc used although there
> are so many AKH transcriptomes/genomes from Hemiptera published and
> only two are mentioned here? In the Figures panels of a to f are not
> correct etc etc. I have not looked carefully at all aspects but the manuscript gives one the impression that it was prematurely send for review. Authors should check carefully English language, prepare a much clearer method section and a coherent and logic Introduction.
>
Round 2
Reviewer 1 Report
The MS was revised in many parts and can be accepted for publication.
As many insect neuropeptides are published in many species, it may be difficult to show the qriginality. However, the effect of BtHTH on feeding behavior is unique if HTH control the appetite and may be useful for controlling pests and plant virus transmission.
In L328-330, it may be better to add the possibility of the inadequate reduction of HTH by RNAi . I corrected as follows:
"The result of mortality of whiteflies in RNAi showed no significant difference in feeding with dsGFP and dsHTH (Figure 2B). This might suggest that RNAi did not decrease the expression level of BtHTH to cause mortality or BtHTH is not indispensable for survival."
Reviewer 2 Report
The authors have significantly improved the manuscript based on the reviewer suggestions and now the manuscript meets the journal standards.
Minor revisions as under-
1. Pl. check lines 99-101 (Gene cloning). I think it is not required here.
2. Primer 3 or 5. Please provide reference
3. Line No. 299-300. Please improve clarity.
4. Please include the full-length and deduced amino acid sequence of BmHTH as Figure S1.
Reviewer 3 Report
The response of the authors to the comments of the referee are very revealing in their "rebuttal" letter:
1. The English is appalling at more than a few places (rewrited instead of rewritten is the most obvious one; but "the sentence making and word usage are recomposed" is also a quite interesting but not really understandable construction).
2. If NCBI database calls a gene or peptide "hypertrehalosaemic" authors should not blindly accept that but use logic and define terms.
The revised manuscript is unfortunately not sufficiently clear by English language, by logic and by science.
To help authors a bit further I will comment on the Introduction but my advice would be: retract the manuscript, put it back into a drawer and READ extensively about the AKH/RPCH family and about phylogenies. Although there are already 13 authors, it is envisaged that the group hires one or two scientific writers who have English as mother tongue, understand the literature on AKH/RPCH and are capable to write a coherent, logical and scientific manuscript for a readership that is not familiar with the topic.
Lets go to the Introduction: Authors have tried to change and have tried to get Hemiptera papers cited but this is all done in a haphazard way without any logic and therefore a non-initiated reader cannot follow this. For example, ref 2 is a very specific publication and cannot be used for a general statement. Ref 1 is a review and is fine here. Lines 38-40: almost all is wrong here: it is corpora cardiaca; it is Locusta migratoria; the reference for this statement is missing; and what do authors mean with "discovered"? Sequenced? By biological assay?
Ref 3 and 4 are totally inappropriate for the sentence in lines 40/41. Ref 3 is "mode of action" and fits later much better, ref 4 could be correct for the previous sentence (it is the first discovery of adipokinetic biological activity). For the more than 60 analogs or bioanalogs (not forms!) of AKHs more recent publications from the Gaede group would be appropriate. Line 44: "variable" what is variable here? residue is missing. The next paragraph, lines 45-51, is quite poor English (what, for example, is nutritional privation condition?), the science is not correct (AKHs regulate not how much energy is consumed; they rather control the level or availability of energetic metabolites to fulfill the metabolic needs of the animal). For this statement the ref 9 and especially 10 are again not correct: your ref 3 would be appropriate here; why cite 10 (Brummer lipase) which is NOT influenced by AKH?).
Next paragraph, lines 52-58: why cite now work on Hemiptera? Because referee wanted it. But not nilly willy! Authors have not introduced the reader that they work with Hemiptera in this paper. The whole build up is not logic.
It would make much more sense if authors would start the Introduction with description of Bemisia and its biology etc. And then mention that genome is known now and one can mine a member of the AKH family. And not call it here already HTH because at this stage the reader does not know what this peptide may be involved in. Then talk a bit general about AKHs and let the reader know what AKH members do in Hemiptera (lots of publications done; a recent review even published in Frontiers of Insect Science) and thereafter pose some scientific questions that this research here wants to answer and what hypotheses are proposed.
Although there are 13 authors and all may be good in handling molecular methods, none is capable of writing a scientific paper in a logical fashion. Mainly because the English is not sufficient to express themselves clearly but there is also a lack of understanding the research topic (role of AKHs and history).
Lets talk a bit on Materials and Methods:
line 88: Authors have never introduced the definition of what a biotype is; and if they "identify" a biotype, why do they not tell us HOW they do that?
line 96-98: Authors use NCBI data base but do not give accession number. How was cDNA synthesized? Was total RNA extracted? From corpora cardiaca?
line 100: Authors have to explain abbreviations in full when they use it the first time, GFP in this case but later also Trans1-T1 etc
line 108: but also previously and later: authors use BTHTH in italics, this should only be done when reference to the GENE is made! Moreover, there is an accepted nomenclature for AKHs; authors use it sometimes (the five letter code) but not consistently, never for Bemta-HTH in this case.
line 113: transmembrane domain. Of an AKH? Authors probably mean of the precursor protein but not the AKH. This lack of detail and real understanding is found more or less throughout the manuscript, is irritating and scientifically wrong.
line 119: this referee does not understand why a beetle (Coccinella) was chosen as outgroup but later (line 202 in Results) it is stated that this Coccinella AKH is "strongly associated".
lines 161 and following: it is not enough to refer to a test kit; authors do not even tell reader WHERE the test kits and reagents are made; where is Solebo situated? Are they available for everyone? Where can they be purchased?
There may be experimental flaws with the starvation experiments, for example. Or authors have omitted to inform reader correctly: it would be helpful to tell the reader how to distinguish between males and females in these small insects (microscopically?). What control was done for the "stress" condition of handling to put the 50 whiteflies into the tube and for being confined under crowded conditions for the whole time? So, maybe not all of the changes are initiated by starvation but by the factors mentioned above.
line 126: give accession number of actin gene sequence
I will not go further into detail. Authors should know by now what to look out for.
There are innumerable sentences where English is so bad that the reader will not understand the meaning. And there are at least 20 to 40 spelling errors or mistakes (in references quite a few).
The citations are mostly not appropriate